# SARS-CoV-2, a Threat to Marine Mammals? A Study from Italian Seawaters

**DOI:** 10.3390/ani11061663

**Published:** 2021-06-03

**Authors:** Tania Audino, Carla Grattarola, Cinzia Centelleghe, Simone Peletto, Federica Giorda, Caterina Lucia Florio, Maria Caramelli, Elena Bozzetta, Sandro Mazzariol, Giovanni Di Guardo, Giancarlo Lauriano, Cristina Casalone

**Affiliations:** 1Istituto Zooprofilattico Sperimentale del Piemonte, Liguria e Valle d’Aosta, 10154 Torino, Italy; tania.audino@izsto.it (T.A.); carla.grattarola@izsto.it (C.G.); simone.peletto@izsto.it (S.P.); federica.giorda@izsto.it (F.G.); caterinalucia.florio@izsto.it (C.L.F.); maria.caramelli@izsto.it (M.C.); elena.bozzetta@izsto.it (E.B.); 2Department of Comparative Biomedicine and Food Science, University of Padua, Legnaro, 35020 Padua, Italy; cinzia.centelleghe@unipd.it (C.C.); sandro.mazzariol@unipd.it (S.M.); 3Institute for Animal Health and Food Safety (IUSA), Veterinary School, University of Las Palmas de Gran Canaria, Las Palmas de Gran Canaria, 35416 Canary Islands, Spain; 4Faculty of Veterinary Medicine, University of Teramo, Strada Provinciale 18 Località Piano d’Accio, 64100 Teramo, Italy; gdiguardo@unite.it; 5Italian National Institute for Environmental Protection and Research (ISPRA), via Vitaliano Brancati 60, 00144 Rome, Italy

**Keywords:** SARS-CoV-2, ACE-2, marine mammals, Italian wastewater management species

## Abstract

**Simple Summary:**

Growing concern exists that SARS-CoV-2, as has already been ascertained for its SARS-CoV and MERS-CoV “predecessors”, originated from an animal “reservoir”, thereafter spilling over into mankind, possibly anticipated by a viral “passage” into a secondary animal host. Within the dramatic SARS-CoV-2 pandemic context, hitherto characterized by over 110 million cases and almost 2,500,000 deaths on a global scale, several domestic and wild animal species have been reported as being susceptible to natural and/or experimental SARS-CoV-2 infection. In this respect, while some marine mammal species are deemed as potentially susceptible to SARS-CoV-2 infection on the basis of the sequence homology of their ACE-2 viral receptor with the human one, this study addresses this critical issue in stranded sea mammal specimens.

**Abstract:**

Zoonotically transmitted coronaviruses were responsible for Severe Acute Respiratory Syndrome Coronavirus-2 (SARS-CoV-2), causing the dramatic Coronavirus Disease-2019 (CoViD-19) pandemic, which affected public health, the economy, and society on a global scale. The impact of the SARS-CoV-2 pandemic permeated into our environment and wildlife as well; in particular, concern has been raised about the viral occurrence and persistence in aquatic and marine ecosystems. The discharge of untreated wastewaters carrying infectious SARS-CoV-2 into natural water systems that are home to sea mammals may have dramatic consequences on vulnerable species. The efficient transmission of coronaviruses raises questions regarding the contributions of virus-receptor interactions. The main receptor of SARS-CoV-2 is Angiotensin Converting Enzyme-2 (ACE-2), serving as a functional receptor for the viral spike (S) protein. This study aimed, through the comparative analysis of the ACE-2 receptor with the human one, at assessing susceptibility to SARS-CoV-2 for different species of marine mammals living in Italian waters. We also determined, by means of immunohistochemistry, ACE-2 receptor localization in the lung tissue from different cetacean species, in order to provide a preliminary characterization of ACE-2 expression in the marine mammal respiratory tracts. Furthermore, to evaluate if and how Italian wastewater management and coastal exposition to extreme weather events may led to susceptible marine mammal populations being exposed to SARS-CoV-2, geomapping data were carried out and overlapped. The results showed the potential SARS-CoV-2 exposure for marine mammals inhabiting Italian coastal waters, putting them at risk when swimming and feeding in specific risk areas. Thus, we highlighted the potential hazard of the reverse zoonotic transmission of SARS-CoV-2 infection, along with its impact on marine mammals regularly inhabiting the Mediterranean Sea, while also stressing the need for appropriate action in order to prevent further damage to specific vulnerable populations.

## 1. Introduction

On 11 March 2020, the World Health Organization (WHO) officially declared the “novel CoronaVirus Disease” (CoViD-19) outbreak, caused by Severe Acute Respiratory Syndrome Coronavirus-2 (SARS-CoV-2), a global pandemic. In Europe, Italy was one of the countries where the earliest cases of CoViD-19 were reported, with 3,962,674 cases, more than 100,000 of which fatal, at the date of our manuscript’s submission on 3 May 2021 (https://www.ecdc.europa.eu/en/geographical-distribution-2019-ncov-cases) (accessed on 3 May 2021). Several works confirmed the presence of SARS-CoV-2 RNA in stools and urine from infected patients [1,2,3], with some reports confirming the presence of both SARS-CoV and SARS-CoV-2 genome fragments in wastewater, sewage sludge, and river waters around the world [4,5,6]. SARS-CoV-2 can remain active for up to 25 days at 5 °C in water sources, based on in vitro studies [7], and can survive in water in a wide range of pH values (3–10) at room temperature [8]. Contaminated water sources can deliver the equivalent of >100 SARS-CoV-2 genome copies with 100 mL or less water in countries with a high prevalence of SARS-CoV-2 infection [7].

These data increase the possibility that wastewater, besides representing a potential, non-invasive early warning tool for monitoring the status and trends of SARS-CoV-2 infection [9,10], may also provide a significant vehicle for spreading this coronavirus [1]. Domestic wastewaters are managed through different treatments, namely: (i) primary (sedimentation); (ii) secondary (biological), which combines aeration tanks with secondary sedimentation to retain the activated sludge; and (iii) tertiary (more stringent), which includes various processes used to further reduce pathogen concentrating and ensure virus-free effluent, e.g., sand filtration and disinfection. The available data, referring to Spain, Italy [1,11], France [9], and Ecuador [12], suggest that nontreated, primary treated and, in some cases, also secondary treated wastewater effluents may represent a risk factor for SARS-CoV-2 transmission [10,11]. These investigations emphasize the limitations of conventional wastewater treatment processes in reducing SARS-CoV-2 RNA with the possible spread in the aquatic environment, including marine waters. Two studies, in fact, have documented the presence of SARS-CoV-2 RNA in an Italian river and a Japanese river, without viral isolation [11,13]. Specifically, the rapid warming reported in the last decades in the Mediterranean Sea basin [14] may occasionally flush untreated or not efficiently treated sewage into rivers and coastal waters when treatment plants reach their capacity and start to fail [15]. Furthermore, it should be noted that extreme weather events (e.g., heavy rainfalls, storms, and floods) associated with global warming, can play a relevant role in coastal pollution [16]. These events could represent a possible risk for susceptible species living close to the shore, such as marine mammals.

These aquatic species, in fact, were recently added to a long list of animal species susceptible to natural and/or experimental SARS-CoV-2 infection [17,18]. In detail, eexperimental infections and binding-affinity assays between the “receptor binding domain” (RBD) of the SARS-CoV-2 spike (S) protein and its receptor, angiotensin-converting enzyme-2 (ACE-2), demonstrate that SARS-CoV-2 has a wide host range because of the high similarity of this protein among difference species [19,20]. Among those animal species showing a higher similarity with the human ACE-2 RBDs [18,21,22], marine mammals seem to have a higher binding efficiency [18] and, for these reasons, infection can be caused by a low viral concentration, such as those likely present in wastewater [23].

The present study aimed to evaluate the susceptibility of marine mammals stranded along the Italian coastlines to SARS-CoV-2 by comparing the amino acid sequences of their ACE-2 receptors and assessing possible menaces to their conservation related to the anthropogenic transmission of SARS-CoV-2. The presence of the ACE-2 receptor molecule in the lung tissues of stranded sea mammals was evaluated by means of immunohistochemistry (IHC), considering the abundance and distribution patterns of the viral receptor (more receptor availability could enhance viral entry into cells); alternatively, ACE-2 may play a protective role against lung (and cardio-vascular) injury through its enzymatic activity [24].

## 2. Materials and Methods

### 2.1. Evaluation of the Susceptibility to SARS-CoV-2 of the Different Species of Marine Mammals in Mediterranean Sea through Comparative ACE-2 Receptor Analysis

#### 2.1.1. ACE-2 Protein Sequences in Marine Mammals’ Analysis

We analyzed the ACE-2 receptor of the marine mammal species residing in the Mediterranean by means of available genomes in the NCBI database. A list of ACE-2 orthologs from Cetacea was downloaded from the NCBI website (https://www.ncbi.nlm.nih.gov/gene/59272/ortholog/?scope=7742, accessed on 15 February 2021). ACE-2 coding DNA sequences were extracted from available or recently sequenced genome assemblies, with the help of a genome alignment tool (BLAST), and the translated protein sequences were checked against human or within family ACE-2 orthologs.

#### 2.1.2. Species Distribution along the Italian Coastline

Several sources on cetacean species’ occurrence and distribution are available for the Mediterranean Sea and for the seas around the Italian peninsula; nonetheless, keeping in mind the aims of this work, we decided to refer to the official updated synthesis available in the IV Reporting of the Habitat Directive (Table 1).

Species distribution and range maps were created using a standard 10 × 10 km ETRS89 grid (projection ETRS LAEA 5210) and a specifically designed “range tool” (http://discomap.eea.europa.eu/App/RangeTool/, accessed on 3 May 2021).

#### 2.1.3. Assessment of Species Susceptibility to SARS-CoV-2 Infection

Of the species listed in Table 1, we examined marine mammal species that encompassed all publicly available reference and scaffold genomes.

For the species lacking genome sequences, ACE-2 primers were designed by identifying conserved regions from multiple alignments of cetacean ACE-2 mRNA sequences (Table 2).

Moreover, for the species in which the genome sequence was lacking, we decided to experimentally determine the ACE-2 receptor sequence based on the species distribution and sample availability.

As striped dolphin represents the most common cetacean in the Mediterranean Sea, and considering the samples availability in the “Centro di Referenza Nazionale per le Indagini Diagnostiche sui Mammiferi Marini Spiaggiati” (C.Re.Di.Ma.) tissue bank, we proceeded with the analyses of this species.

We used the set of rules developed by Damas et al., 2020 [25], for predicting the likelihood of the SARS-CoV-2 S protein binding to cetacean ACE-2. This study identified the ACE-2 amino acid residues involved in binding to the SARS-CoV-2 S protein that were previously reported to be critical for the effective binding of ACE-2 to SARS-CoV-2 S protein’s RBD. These residues include S19, Q24, T27, F28, D30, K31, H34, E35, E37, D38, Y41, Q42, L45, L79, M82, Y83, N330, K353, G354, D355, R357, and R393. The known human ACE-2 RBD glycosylation sites N53, N90, and N322 were also included in the analyzed amino acid residue set.

Regions of the human ACE-2 protein sequence interacting with the SARS-CoV-2 S protein were highlighted and compared with the amino acid composition of the protein sequences of marine mammals selected for the study (Table 3).

Species were classified in one of five categories, namely: very high, high, medium, low, or very low susceptibility.

#### 2.1.4. Molecular Analyses (PCR and Sequencing)

Striped dolphin brain and lung samples were submitted to total RNA extraction using the TissueLyser II (QIAGEN, Hilden, Germany) and highspeed shaking in Eppendorf tubes with stainless steel beads (5 mm diameter, QIAGEN, Hilden, Germany). Homogenates were centrifuged at 14,000 rpm for 3 min to remove tissue debris. Supernatants were used for RNA extraction with the AllPrep DNA/RNA Mini kit (QIAGEN, Hilden, Germany) according to the manufacturer’s instructions. RNA was eluted in a final volume of 100 μL of elution buffer and stored at −80 °C until analyzed.

The total RNA was reverse transcribed into cDNA using the High-Capacity cDNA Reverse Transcription kit (Thermo Fisher Scientific, Waltham, MA, USA), according to the manufacturer’s protocol.

ACE-2 primers were designed using Primer3 software from multiple alignments of cetacean ACE-2 mRNA sequences. The ACE-2 conserved regions were identified flanking the nucleotide triplets of interest. Two primer pairs were designed, amplifying two ACE-2 gene fragments (Table 2).

The reaction system used for PCR amplification was 50 μL, consisting of ACE2 primers 2.50 μL, 1× Buffer 2.50 μL, MgCl2 2 μL, dNTPs 0.50 μL, AmpliTaq Gold 0.20 μL, cDNA 2.50 μL, and H2O 12.30 μL.

The reaction was performed with the following PCR conditions: 5 min at 95 °C; followed by 40/48 cycles of 30 s at 94 °C, 30 s at 55 °C, and 30 s at 72 °C; and 7 min at 72 °C.

The amplification products were analyzed by electrophoresis in a GelRed (Biotium, Fremont, CA, USA) staining 2% agarose gel, visualized under UV light transillumination (Gel-Doc UV transilluminator system Bio-Rad, Hercules, CA, USA) and were then identified by their molecular weights.

Amplicons were purified using the NucleoSpin Gel and PCR Clean-up kit (Macherey-Nagel, Dueren, Germany), and were submitted to a cycle-sequencing reaction using the BigDye Terminator v.3.1 Cycle Sequencing kit (Thermo Fisher Scientific, Whaltam, MA, USA).

The sequences were manually inspected using the Sequencing Analysis v. 5.2 software. A dataset in fasta format was created using the newly generated sequences and ACE-2 homologue sequences available in GenBank. The dataset was aligned using BioEdit v7.2.5 software for the identification of the ACE-2 amino acid residues of interest.

#### 2.1.5. Cross-Referencing of Conservation Status and Susceptibility

We cross-referenced the International Union for Conservation of Nature (IUCN) Red list of Threatened Species (https://www.iucnredlist.org; accessed on 31 July 2020) to better understand which at risk species could also be susceptible to the virus. For each species, we looked at the Global, Mediterranean, and Italian () assessment in order to determine their IUCN Red List Category. The Mediterranean IUCN status is under review (Table 4).

### 2.2. Geo-Mapping of Italian Wastewater Plants, Stretches of Coast and Species at Risk

To identify high-risk areas for SARS-CoV-2 viral spillover in Italy, we overlaid geo-mapping data of Italian wastewater plants with local marine mammal population data.

In order to assess the potential viral infection level of water bodies entering the sea, we firstly considered quality sewage treatment procedures and the location of the Italian wastewater plants. Subsequently, we took into account the coastal areas more frequently exposed to the effects of global warming (e.g., heavy rainfalls, storms, and floods) and consequently at high risk of flush untreated or not efficiently treated sewage into rivers and coastal water.

For the first purpose, we referred to the European Environmental Agency thematic page (https://www.eea.europa.eu/themes/water/european-waters/water-use-and-environmental-pressures/uwwtd) (accessed on 3 May 2021), focusing on our country. We used the urban wastewater treatment map based on provisional data on the implementation of the Urban WasteWater Treatment Directive (UWWTD) in EU Countries from 2018, which were reported by the concerned countries in 2020.

To search for data on wastewater plants with poor quality sewage treatment procedures, we considered both the urban wastewater treatment pathways in big cities (agglomerations ≥ 150,000 population equivalent (p.e.) and the urban wastewater treatment plants from agglomerations ≥ 2000 p.e.).

Regarding the type of treatment, in both cases, we focused on primary and secondary treatment plants, which gave no guarantees of virus inactivation, in order to locate first level risk areas.

In order to identify risk areas at a higher level of detail, we focused on treatment plants (reported for agglomerations ≥ 2000 p.e.) with more stringent treatments (e.g., disinfection, sand filtration, and other), selecting the location of plants that applied a type of sanitation other than disinfection (chlorination, UV, or ozonisation).

For the cartographic return/mapping, to build the first layer, we considered the sites most at risk overall.

To identify the coastal areas more involved in extreme weather events, we referred to the 2018 edition of the report prepared by the Italian Institute for Environmental Protection and Research (ISPRA) (https://www.isprambiente.gov.it/files2018/pubblicazioni/rapporti/rapporto-dissesto-idrogeologico/infografica_Rapporto_Dissesto_idrogeologico_2018.pdf), providing an updated overview on landslide and flood hazards throughout the country.

The distribution maps were overlaid with data of the wastewater treatment plants and on the coastal areas potentially more involved in extreme weather events.

Using this approach, we were able to determine each geographic area and its surroundings, where wastewater effluents and overflows may pose marine mammals at risk from anthropogenic SARS-CoV-2 transmission.

### 2.3. Immunohistochemistry

Previous studies on SARS-CoV-2 have shown that ACE-2 is the receptor through which the virus enters cells. In order to understand the patterns of ACE-2 protein expression in the lung tissue and if this is relevant to the possibility of acquiring and developing SARS-CoV-2 infection by cetaceans, we carried out ad hoc IHC investigations on different species of cetaceans (Table 5), based on pulmonary tissue sample availability at the Mediterranean Marine Mammal Tissue Bank (MMMTB) of the University of Padua (Legnaro, Padua, Italy). IHC staining for ACE-2 was performed as outlined below.

In order to evaluate the IR patterns of ACE-2, lung tissue sections from different species of cetaceans, previously formalin-fixed and paraffin embedded, were sectioned and hydrated through xylenes; endogenous peroxidase was blocked using a 3% hydrogen peroxide solution. Heat-induced antigen retrieval was performed using a citrate buffer bath (PH 6.1) at 97 °C for 15 min. After cooling to room temperature, the sections were incubated with a blocking serum (VECTASTAIN ABC Kit, pk-4001, Vector Laboratories, Burlingame, United States) for 20 min before being incubated overnight with the primary anti-ACE-2 antibody (a polyclonal antibody raised in rabbits and diluted 1:2000, ab15348 Abcam, Cambridge, UK) at 4 °C in Tris-Buffered Saline (TBS) containing Tween. The next day, after washing with TBS buffer, sections were first incubated with the secondary antibody for 30 min, then with the prepared VECTASTAIN ABC Reagent; subsequently, the final detection was carried out with diaminobenzidine (DAB, Dako K3468, Abcam, Cambridge, UK) as chromogen for 5 min. The sections were then counterstained with Mayer’s hematoxylin for a better visualization of the tissue morphology. Adequate positive and blank control tissues were utilized in each run.

## 3. Results

### 3.1. Susceptibility of the Marine Mammal Species Examined to SARS-CoV-2

Our analyses suggested which species of Mediterranean Sea mammals living in Italian waters are most susceptible to SARS-CoV-2 infection by assessing the primary sequence homology level of their ACE-2 receptor with the human one.

At present, in the whole Mediterranean Sea, 12 marine mammal species are listed as being present regularly: 11 Cetaceans (Balaenoptera physalus, Physeter macrocephalus, Ziphius cavirostris, Globicephala melas, Orcinus orca, Grampus griseus, Tursiops truncatus, Stenella coeruleoalba, Delphinus delphis Delphinus capensis, and Steno bredanensis) and 1 Pinniped (Monachus monachus). Moreover, the presence of other three species is considered to be occasional (Balaenoptera acutorostrata, Pseudorca crassidens, and Megaptera novaengliae; Table 1).

Out of the 15 aforementioned species, the ACE-2 amino acid sequence was available in the NCBI database for 8 of them (*B. physalus*, *P. macrocephalus*, *Z. cavirostris*, *G. melas*, *O. orca*, *T. truncatus*, *B. acutorostrata*, and *M. novaengliae*).

Among the species lacking ACE-2 sequence-related data, no information was publicly available before our study on *S. coeruleoalba* (a species commonly occurring around the world in temperate waters), for which these data were obtained experimentally.

Based on the 25 amino acid residues selected, the sequences collected from the 9 concerned species, compared with the corresponding ones from the human ACE-2 receptor, are shown in Table 3. Based on the primary structure homology level of their ACE-2 molecule with the human one, the majority of *Cetacea* species analyzed (7/9 species) were predicted to be highly susceptible to SARS-CoV-2 infection (*B. physalus*, *G. melas*, *T. truncates*, *O. orca*, *S. coeruleoalba*, *B. acutorostrata*, and *M. novaengliae*), with the remaining two species *(P. microcephalus* and *Z. cavirostris)* being predicted as “medium susceptibility” species (Table 6).

In more detail, in all of the examined marine mammal genomes, the binding residues D30 and M82 of ACE 2 were both mutated; however, these two mutations did not apparently exert a prominent destabilizing effect on the affinity of SARS-CoV-2 S protein’s RBD to ACE-2 [25], thus not affecting the predicted susceptibility of the concerned cetacean species to SARS-CoV-2. Instead, the L79 and M82 mutations detected in *P. macrocephalus* and in *Z. cavirostris* impacted the binding affinity of the virus to ACE-2, thereby decreasing the predicted susceptibility of the latter two species to SARS-CoV-2 infection (Table 6).

The IUCN status for the investigated species is available at a different geographic range.

Fin whale (*B. physalus*), common bottlenose (*T. truncatus),* and striped dolphins (*S. coeruleoalba*) are all vulnerable at the Mediterranean scale, but display a different assessment at the Italian range (near threatened and least concern, respectively).

Pilot whale (*G. melas*) and Cuvier’s beaked whale *(Z. cavirostris*) are data deficient at both scales, with no differences between Mediterranean and Italian ranges being reported for the status of the sperm whale (endangered).

Finally, both minke (*B. acutorostrata*) and humpback (*M. novaengliae*) whales have not been assessed for the Mediterranean Sea, given their irregular occurrence in the Basin.

The results of the comparison between the IUCN conservation status and SARS-CoV-2 susceptibility for the 9 cetacean species investigated herein are summarized in Table 6.

### 3.2. Wastewater Treatment Plants Conditions, Hydrogeological Vulnerability, and Risk to Marine Mammal Species in Italian Seas

The identification of wastewater treatment plants and the assessment of the municipality contribution in terms of untreated or not sufficient treated wastewater discharge into natural water systems can help to predict the potential hotpots for a spillover event.

In Italy, wastewaters are treated by primary, secondary, and more stringent treatment plants (WWTPs).

Considering the potential risk of plants located in big municipalities (agglomerations ≥ 150,000 p.e.), we found that the big municipalities of the northern and central regions of our Country rely mostly on more stringent treatment plants, with a few exceptions (Figure A1), represented in particular by the city of Trieste, in north-eastern Italy, with a predominance of primary treatment plants, as well as by the city of Pescara, in central Italy, with a majority of secondary treatment plants.

By contrast, the big municipalities in Southern Italy have a large number of secondary treatment plants and, in some cities (Catania and Cosenza), a rate of wastewaters not collected in sewerage systems is reported, along with a rate not submitted to treatments or characterized by a lack of information (Catania).

Considering the potential release of SARS-CoV-2 (as well as of other oro-fecally transmitted viral and non-viral agents) into the effluents of primary and secondary plants upstream of relevant rivers, we identified the North Adriatic Sea and the Ligurian Sea as basins being at risk for the potential introduction of effluents into the Po and Arno rivers. Moreover, considering the location of some plants with low sanitization guarantees close to the shores of central and southern Italy, we recognized an additional risk factor for the Central Adriatic Sea, the Northern and Southern Ionian Sea, and the South-Eastern Tyrrhenian Sea.

Taking into account the urban wastewater treatment plants from agglomerations ≥ 2000 p.e (UWWTD) and focusing in particular on their location along the entire shoreline, we identified that some coastal areas, with a prevalence of primary and secondary plants, appearred at high risk, as well as the water basins on which they overlook (Ligurian Sea, Central Adriatic Sea, South-Eastern Tyrrhenian Sea, Northern and Southern Ionian Sea, and Sicilian Channel; Figure A2).

In more detail, focusing on plants with more stringent treatments, such as disinfection (chlorination, UV, and ozonisation), sand filtration, micro filtration (e.g., membrane filtration), or other types of unspecified additional treatment, we considered the potential risk of UWWTD not applying the disinfection, along with their location close to the shores, and we recognized at risk basins being those of the Northern Adriatic Sea, the Central Adriatic Sea, and the Central Tyrrhenian Sea (Figure A3).

Moreover, considering the coastal areas more exposed to landslides and flood hazards, we identified the coastline of the Ligurian Sea and of the Central and South-Eastern Tyrrhenian Sea as sites potentially at risk for occasional over-flooding of sewage treatment plants and the release of wastewater not fully treated, which may transfer, at their turn, microbial pathogens from distant sources to coastal waters. (Figure A4).

Overall, we identified some high-risk areas for a potential SARS-CoV-2 spillover to sea mammals, as summarized in Figure 1, which correspond to (clockwise) the Northern Adriatic and Central Adriatic Seas, the Northern and Southern Ionian Sea, the Strait of Sicily, and the whole Tyrrhenian and the Ligurian Seas.

The geographic distribution of the marine mammal species under study, along with the risky areas for SARS-CoV-2 spillover are shown in Figure 1. The risky areas have been inferred through a combination of UWWT maps (https://www.eea.europa.eu/themes/water/european-waters/water-use-and-environmental-pressures/uwwtd, (accessed on 3 May 2021) coupled with landslide and flood hazards (https://www.isprambiente.gov.it/files2018/pubblicazioni/rapporti/rapporto-dissesto idrogeologico/Rapporto_Dissesto_Idrogeologico_ISPRA_287_2018_Web.pdf) (accessed on 3 May 2021). The concerned species’ distribution was derived from the fourth Italian Report to the Habitat Directive.

The risk areas as indicated by the study host several cetacean species, as shown in Figure 1. The map was created considering the data available in the IV Italian Report to the Habitat Directive; given the irregular status of both minke and humpback whales, coupled with the occurrence of pilot and killer whales, these species were not reported on the map.

From the map the overlap between the inferred risky areas with the main distribution of the species is clear; it is noteworthy that the concerned map should be viewed as a synthesis and a generalization of the available data on cetaceans’ distribution in the seas around the Italian peninsula. In this respect, more detailed data are available, even on a small-scale base, and these would increase the species range displayed in Figure 1. In such a context, a suitable area for the fin whale should be much more extended on a seasonal basis to the entire Ligurian Sea, representing a well know summer fishing ground for this species [26].

The distribution of the *P. macrocephalus* overlaps the North Tyrrhenian and the Ligurian Seas; *B. physalus* distribution overlaps the North Tyrrhenian Sea, the Ligurian Sea, and the Strait of Sicily; and *T. truncatus* distribution overlaps the Ligurian sea, the whole Tyrrhenian Sea area, the Strait of Sicily, and the North and Central Adriatic Sea.

*S. coeruleoalba* distribution overlaps the Ligurian sea, the whole Tyrrhenian Sea area, the Strait of Sicily, and the Ionian Sea.

*Z. cavirostris* overlaps the North Tyrrhenian Sea.

*G. griseus* distribution overlaps the Ligurian Sea, the Strait of Sicily, and the Ionian Sea.

### 3.3. Immunohistochemical (IHC) Characterization and Pulmonary Location of ACE-2

Because the ACE2 protein shares its role as a cellular receptor of SARS-CoV-2, plenty of research has been focused on the ACE2 expression in various organs, especially in the respiratory system, which can be the entrance point for SARS-CoV-2.

ACE-2 does not appear to be the only receptor for SARS-CoV-2; additional cells receptors such as “neuropilin-1” (NP-1) [27] and (HDL)-cholesterol scavenger receptor B-type 1 [28] have been reported to be either co-expressed or not concurrently expressed with ACE-2, and are involved in viral host cell interaction. Furthermore, the ACE-2 molecule is known to serve as the main cell receptor for SARS-CoV-2.

The ACE-2 receptor has already been described and IHC characterized in numerous mammalian species; unlike terrestrial mammals, the lungs of cetaceans undergo anatomical and physiological adaptations that facilitate extended breath-holding during dives. For these reasons, and to investigate the presence and location of the ACE-2 receptor in marine mammals, the lung tissue of seven cetacean species were IHC tested (Table 5).

The expressions of ACE2 were visualized by immunohistochemistry on paraffin-embedded lung tissue samples of different species of cetacean; as shown in Figure 2 and Figure 3, the brown areas indicate ACE2+ staining, corresponding to alveolar and bronchial epithelium.

All of the examined species showed lung tissue immunolabelling for ACE-2 antibody. In particular, the most remarkable finding of IHC investigations was the detection of the ACE-2 expression on the surface of alveolar (type I pneumocytes) and bronchiolar epithelial cells. Furthermore, ACE-2-specific immunoreactivity (IR) was observed in endoalveolar macrophages, as well as in the endothelium and in the smooth muscle cells of pulmonary vessels (Figure 2 and Figure 3).

The results of this study should be considered as a preliminary starting point for further studies.

## 4. Discussion

In general, the conservation status of Mediterranean cetaceans is under pressure from anthropogenic disturbance, and the populations of the species highlighted in the study are greatly impacted by several sources of human activities. Incidental mortality in fishing operations, collisions with ships, the ingestion of debris, seismic surveys, naval exercises, chemical pollution, and viral infections are just some of the threats faced by the cetaceans inhabiting the Mediterranean Sea Basin [29,30,31,32]. Most of the aforementioned factors can have a direct effect on the individuals, while indirect effects on the medium long-term period are represented by general habitat degradation.

Marine mammals may be exposed to environmental stressors such as chemical pollutants, harmful algal biotoxins, and emerging or resurging pathogens; this phenomenon may be related to complex factors such as climate change, toxins, and immunosuppression, with coastal marine mammals particularly at risk as many inhabit an environment more affected by human activity [33]. Besides threats of an anthropic origin menacing their conservation, infectious diseases represent a global issue, in particular morbilliviruses [34,35]. These RNA viruses, currently endemic in the Mediterranean, showed an apparently increased tendency towards cross species infection in recent years, causing epidemic events in different species through spillover events. Phylogenetic analyses support the idea of a common ancestor of morbillivirus affecting aquatic animals, namely cetacean morbillivirus (CeMV), with those reported in terrestrial ones and recent reports of this virus in more terrestrial animals (i.e., otters and seals) suggest a link back to land for this virus [36].

The impact of humans on wildlife during a global pandemic may include the potential transmission of a novel virus to susceptible animals [18]. In our study, we identified species of marine mammals living along the Italian coastline and we evaluated the conservation of the SARS-CoV-2 viral ACE-2 receptor across species.

It has been shown how ACE-2 variability explains why certain species are susceptible to SARS-CoV-2, while others are not [37]; ACE-2, an extracellular peptidase originally characterized as a SARS-CoV receptor, has been subsequently identified as the main SARS-CoV-2 receptor. The S protein binding region is located in the ACE-2 catalytic [38], with some amino acid residues at a particular position in the human ACE-2 sequence playing a crucial role in virus–host cell interactions [38,39]; these binding residues determine the degree of susceptibility, thus likely representing the main drivers for cross-species transmission [39].

Our analyses revealed that a group of closely related cetaceans (*O. orca*, *G. melas*, *T. truncates,* and *S. coeruleoalba*) is predicted to be potential highly susceptible to the virus, hence is at potential risk of acquiring SARS-CoV-2 infection whenever exposed to the viral pathogen. One of the main aims of this study was to evaluate ACE-2 abundance and distribution, by means of IHC, in cetacean lungs, thereby supporting our parallel and comparative investigations on ACE-2 amino acid sequences, aimed at assessing the susceptibility of marine mammals to SARS-CoV-2 infection; in this respect, IHC analyses could identify a number of potential routes of infection for SARS-CoV-2, along with the viral spread and replication sites throughout the body. Given the IR patterns found, the expression of ACE-2 in the macrophages and their role in antiviral defense mechanisms (as in the case of SARS-CoV-2 infection) should be emphasized; Abassi et al. have hypothesized that while lung macrophages play an important role in antiviral defense mechanisms, they could also serve as a “Trojan horse” for SARS-CoV-2, thus enabling viral anchoring within the pulmonary parenchyma. A variable expression of ACE-2 on the macrophages among individuals might also govern the severity of SARS-CoV-2 infection [40], although additional studies are required. Our findings suggest that the ACE-2 expression can vary between different lung regions and between individuals (in particular, if they belong to different species).

Reports of pathogens of terrestrial origins in marine mammals have already been observed, as for the case of *Toxoplasma gondii*, *Salmonella typhimurium*, and *Listeria monocytogenes*, and often exposure to untreated wastewater was deemed to be the possible source [41,42].

As with other fecal pathogens, SARS-CoV-2 is transferred through the sewage system, thereby gaining access into wastewater treatment plants, and/or, more generally, the aquatic environment.

In this respect, Italy is characterized by heterogeneous geographic systems, with sea surface waters surrounding the Italian peninsula. The spaces and distances granted to the hydrographic network by the mountains and the sea are mostly very modest, making the territory particularly exposed and vulnerable to alluvial events, known as sudden floods or flash floods, often triggered by short and intense weather phenomena (https://www.isprambiente.gov.it/files2018/pubblicazioni/rapporti/rapporto-dissesto idrogeologico/Rapporto_Dissesto_Idrogeologico_ISPRA_287_2018_Web.pdf) (accessed on 3 May 2021). Compared with the unpredictability of flood events, there is still a sort of repetition in the occurrence of the events themselves, and some portions of our national territory, because of the morphological characteristics and use of soil, are configured as hydrological hazard-prone areas, including coastal areas.

The present study identified some high-risk areas where insufficient wastewater treatment may occur in the vicinity of marine mammals, putting them at risk for infection by a fecally transmitted zoonotic pathogen like SARS-CoV-2 when they swim and feed. It is clear that several data and information are still missing regarding the survival time in the marine environment and the effects of marine currents and dilution factors, which can lower the possibility of infection. Furthermore, it should be noted that SARS-CoV-2 has been found in wastewater, there are no data on its actual possibility of survival and dispersion in seawater.

Analyzing wastewater management practices in Italy, we identified that some wastewater treatment plants in the vicinity of marine mammals utilize tertiary treatment, which rules out the possibility of virus exposure in these areas. However, there are locations (the coastline of the Ligurian Sea and of the Central and South-East Thyrrenian Sea) that border marine mammal populations that are at risk for occasionally over-flooding in sewage treatment plants, resulting in the release of wastewater that has not fully been treated. The number of areas identified as being at risk could be underestimated, as we only considered areas characterized by inadequate treatment, while in some cases, other coexisting conditions, like problematic sewage overflow or pipe exfiltration, represent a risk. Overall, considering all of these factors, we identified some high-risk areas for a potential virus spillover (Figure 1), which corresponded to the North Adriatic Sea, Ligurian Sea, Central Adriatic Sea, North and South Ionian Sea, Central Tyrrhenian Sea, South-East Thyrrenian Sea, and the Sicilian Channel.

The impact of a possible viral spillover could have on coastal marine mammal communities remains to be determined. Our study, in fact, confirmed a high susceptibility of sea mammals to SARS-CoV-2 infection based on the receptor homology, as reported in previous studies [18]. The effects of SARS-CoV-2 on marine mammals are currently unknown, although coronavirus infections have been reported in marine mammals prior to the CoViD-19 pandemic [43]. Infections with other coronaviruses are indeed recognized as a cause of liver and lung disease [44], while gammacoronaviruses were retrieved from the fecal samples of three Indo-Pacific bottlenose dolphins (*Tursiops aduncus*).

As many cetacean species are social, such as *Tursiops truncatus* (bottlenose dolphin) and *Stenella coeruleoalba* (striped dolphin), their high susceptibility to SARS-CoV-2 suggests that their populations could be especially vulnerable to intra-species transmission of this novel coronavirus. Among the species with a high susceptibility to SARS-CoV-2 infection, the common bottlenose dolphin could result in being the most impacted cetacean, given its distribution along the continental shelf and along the entire Italian coastline. Moreover, the behavior and the size of the pods of the species could play an important role in the spread of the virus among the specimens in the pod; both striped and common bottlenose dolphins are gregarious species and this ecological feature may facilitate the SARS-CoV-2 spread among the animals through their close interactions.

Along the entire Italian coast, the most common species is *T. truncatus* [45]; the species is the only regular one in the Northern Adriatic Sea [46], and is also distributed in the Ligurian Sea, the whole Tyrrhenian Sea [47,48], the Sicily Strait, and the Ionian Sea. As *T. truncatus* is an “inshore” species, the risk of acquiring SARS-CoV-2 infection, which like many others is characterized by a “land-to-sea” transmission eco-epidemiological pathway, appears to be greater than for “offshore” species. This would make bottlenose dolphins among the most reliable “sentinels” as candidates for the “early” detection of SARS-CoV-2 infection in marine mammals. *S. coeruleoalba*, the most abundant species in the Mediterranean Sea [49], displays an offshore distribution [50] and is regular in the Southern Adriatic Sea, Ionian Sea including the Sicily Strait, all of the Tyrrhenian Sea, and the Ligurian Sea, along with *B. physalus* [26]. The fin whale distribution ranges from north to south feeding grounds in the summer and winter, respectively, as it has been described by satellite telemetry [26] and boat-based observations [51].

Even if a decline in the population of *D. delphis*, has been reported for the Mediterranean Sea [52], the species is still regularly observed in the Tyrrhenian Sea.

Infection susceptibility can hinder the conservation status of the species; indeed, the fin whale is listed as vulnerable in the IUCN (Mediterranean and Italian range), as a population abundance decline has been inferred in the last years. The decline is a major concern for the Mediterranean sub population [26]; hence, epidemic disease, as already documented for morbilliviral infection [53], would represent a plague. A worst situation would be expected for the critically endangered killer whale, given the extremely low number of individuals left in the subpopulation confined in the Gibraltar Strait [54]. Even if the analysis for the sperm whale revealed a medium susceptibility, the low number of individuals within the population and the Endangered IUCN Mediterranean and Italian status foresee a high-risk degree for such species.

## 5. Conclusions

By highlighting the vulnerability of marine mammals to SARS-CoV-2 infection, the scientific community hopes to shape policy decisions regarding wastewater management around the world in order to help protect at-risk marine mammal species that may be exposed to this coronavirus.

Regarding the Italian situation, additional measures of treating wastewater in the areas where they are insufficient or inadequate (Ligurian Sea, Central Adriatic Sea, South-Eastern Tyrrhenian Sea, Northern and Southern Ionian Sea, and Sicilian Channel) would help protect the species and reduce the probability of them being exposed to SARS-CoV-2.

In summary, there is an urgent need to increase SARS-CoV-2 infection’s surveillance in free-ranging cetaceans, and also to screen stranded specimens for such infection, particularly in cases of mass stranding or unusual behavior.

Further studies are needed in cetaceans to provide a more in-depth insight into SARS-CoV-2 susceptibility, also in relation to the escalating levels of anthropogenic stressors to which they are exposed.

Considering the economic costs of the present pandemic, greater efforts to improve wastewater treatment, specifically the removal or inactivation of viral contaminants on a global scale, should be regarded as a priority.

This is made even more urgent by the new SARS-CoV-2 variants, which have emerged in the United Kingdom, as well as in Brazil and South Africa, with the S protein’s mutations carried by them leading to greater viral transmission/transmissibility rates and, possibly, also to increased pathogenicity.

These variants, which are rapidly spreading around the world, have also been isolated and sequenced on the Italian territory; in this respect, viral genome sequencing from wastewater can detect new SARS-CoV-2 variants before they are detected in infected patients. Consequently, analyzing sewage and run-off waters can help detect and track the spread of new SARS-CoV-2 variants posing a risk to human health and, potentially, also to the health and conservation status of wild animal species and populations, including sea mammals.

## Figures and Tables

**Figure 1 animals-11-01663-f001:**
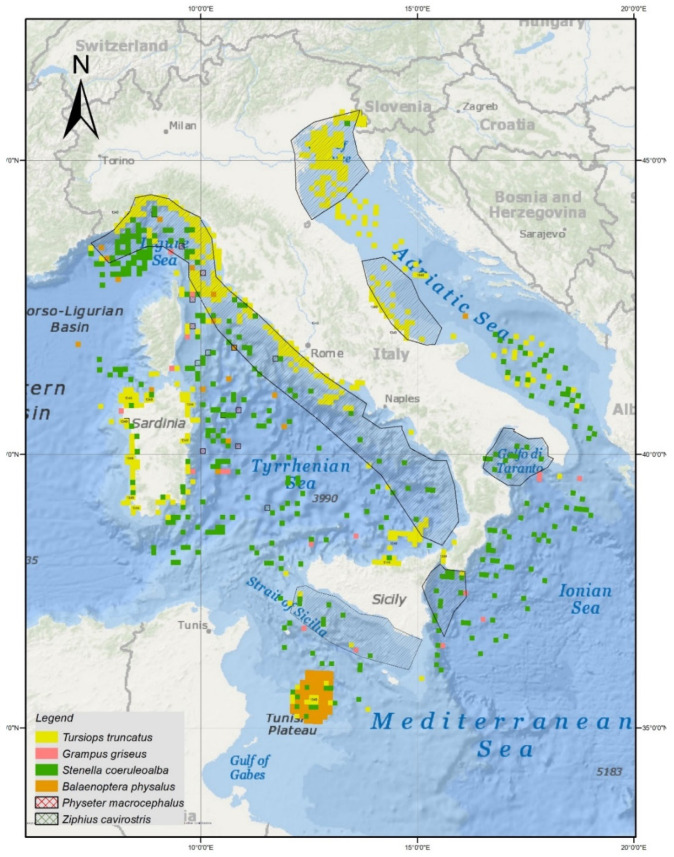
Species and risky areas.

**Figure 2 animals-11-01663-f002:**
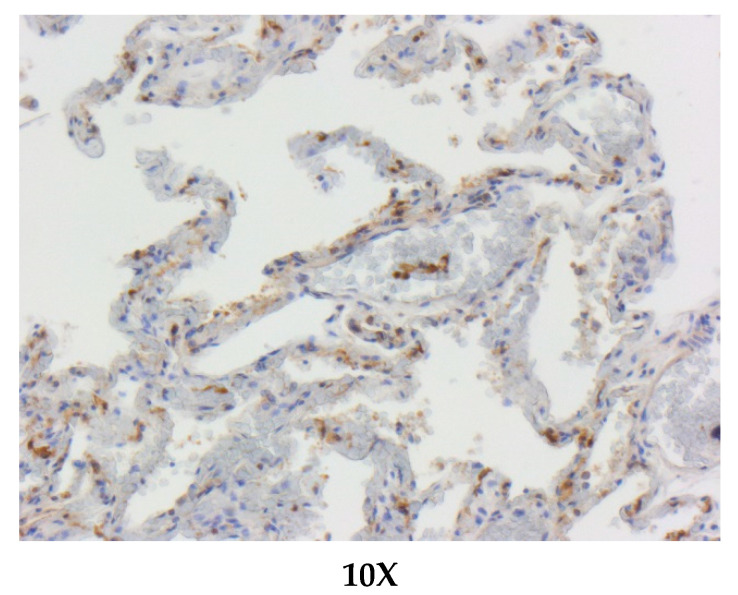
ACE-2 immunohistochemical staining in the lungs of a bottlenose dolphin (*Tursiops truncatus*). Positive immunoreactivity within type I pneumocytes from the alveolar respiratory epithelium and in alveolar macrophages. ACE-2 immunohistochemistry, Mayer’s hematoxylin counterstain, and 10× objectives

**Figure 3 animals-11-01663-f003:**
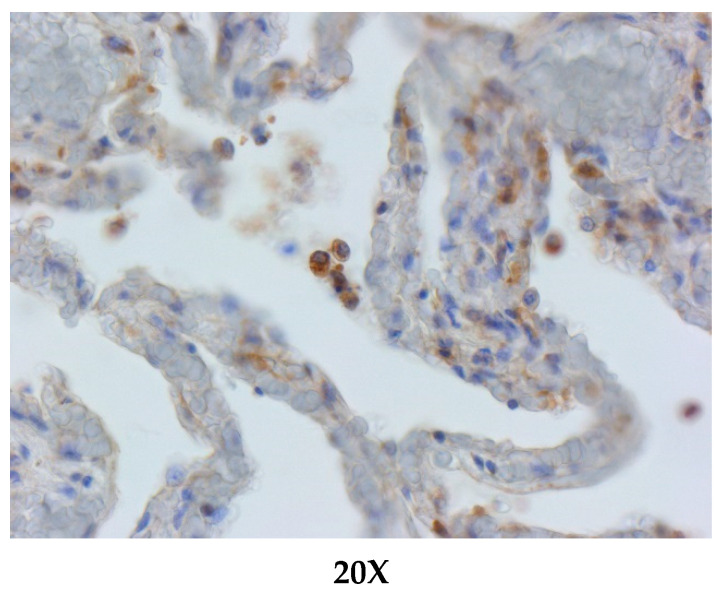
ACE-2 immunohistochemical staining in the lungs of a bottlenose dolphin (*Tursiops truncatus*). Positive immunoreactivity within type I pneumocytes from the alveolar respiratory epithelium and in alveolar macrophages. ACE-2 immunohistochemistry, Mayer’s hematoxylin counterstain, and 20× objectives

**Table 1 animals-11-01663-t001:** Marine mammal species inhabiting Italian seas.

Present Regularly	Occasional
**Cetacea**	
*Balaenoptera physalus*	*Balaenoptera acutorostrata*
*Physeter macrocephalus*	*Pseudorca crassidens*
*Ziphius cavirostris*	*Megaptera novaengliae*
*Globicephala melas*	
*Orcinus orca*	
*Grampus griseus*	
*Tursiops truncates*	
*Stenella coeruleoalba*	
*Delphinus delphis*	
*Delphinus capensis*	
*Steno bredanensis*	
**Pinnipedia**	
*Monachus monachus*	

**Table 2 animals-11-01663-t002:** PCR assays for angiotensin-converting enzyme-2 (ACE-2) characterization in marine mammals and related primers.

Primer	Nucleotide Sequence (5′-3′)	Amplicon Size (bp)
CetACE2_19-90 F	CTCCTTCTCAGCCTTGTTGC	285 bp
CetACE2_19-90 R	CTGAAGGRCCTGCAATTGAC	
CetACE2_322-393 F	AGTCCTGGGATGCAAAGAGG	492 bp
CetACE2_322-393 R	AGACCATTCYCCTCCACTTT	

**Table 3 animals-11-01663-t003:** Primary sequences of ACE-2 from marine mammals and their comparison with human ACE-2.

Sequence Position Number	19	24	27	28	30	31	34	35	37	38	41	42	45	53	79	82	83	90	322	330	353	354	355	357	393
*Homo* *sapiens*	S	Q	T	F	D	K	H	E	E	D	Y	Q	L	N	L	M	Y	N	N	N	K	G	D	R	R
*Balaenoptera physalus*	S	Q	T	F	Q	K	H	E	E	D	H	E	L	N	I	T	Y	N	N	N	K	G	D	R	R
*Physeter macrocephalus*	S	Q	T	F	Q	K	H	E	E	D	Y	Q	L	N	T	T	Y	N	N	N	K	G	D	R	R
*Ziphius cavirostris*	P	K	T	F	Q	K	H	E	E	D	Y	Q	L	N	T	T	Y	N	N	N	K	G	D	R	R
*Globicephala melas*	S	R	T	F	Q	K	R	E	E	D	Y	Q	L	N	I	T	Y	N	N	N	K	G	D	R	R
*Tursiops truncatus*	S	R	T	F	Q	K	R	E	E	D	Y	Q	L	N	I	T	Y	N	N	N	K	G	D	R	R
*Stenella coeruleoalba*	S	R	T	F	Q	K	R	E	E	D	Y	Q	L	N	I	T	Y	N	N	N	K	G	D	R	R
*Balaenoptera acutorostrata*	S	Q	T	F	Q	K	H	E	E	D	Y	R	L	N	I	T	Y	N	N	N	K	G	D	R	R
*Orcinus orca*	S	R	T	F	Q	K	R	E	E	D	Y	Q	L	N	I	T	Y	N	N	N	K	G	D	R	R
*Megaptera novaengliae*	S	Q	T	F	Q	K	R	E	E	D	Y	E	.	N	.	T	Y	N	N	N	K	G	D	R	R
Color legend:.																									
Non-conservative replacement																									
Conservative replacement																									

**Table 4 animals-11-01663-t004:** IUCN status of the marine mammal species under investigation according to geographic range.

Species	IUCN Status
*Balaenoptera acutorostrata* (Minke whale)	LC (gl, 2018)
*Megaptera novaeangliae* (Humpbeak whale)	LC (gl, 2018)
*Orcinus orca* (Killer whale)	CE (Strait of Gibraltar subpopulation, 2019) DD (gl, 2017)
*Pseudorca crassidens* (False killer whale)	NT (gl, 2018)
*Steno bredanensis* (rough-toothed dolphin)	LC (gl, 2018)
*Delphinus delphis* (short beaked common dolphin)	EN (it, 2013), EN (med, 2003), LC (gl, 2008)
*Delphinus capensis* (long beaked common dolphin)	EN (it, 2013), EN (med, 2003), LC (gl, 2008)
*Physeter macrocephalus* (Sperm whale)	EN (it, 2013), EN (med, 2006), VU (gl, 2008)
*Balaenoptera physalus* (Fin whale)	VU (it, 2013), VU (med, 2010), VU (gl, 2018)
*Tursiops truncatus* (Common bottlenose dolphin)	NT (it, 2013), VU (med, 2009), LC (gl, 2018)
*Stenella coeruleoalba* (Striped dolphin)	LC (it, 2013), VU (med, 2010), LC (gl, 2018)
*Globicephala melas* (Pilot whale)	DD (it, 2013), DD (med, 2010), LC (gl, 2018)
*Grampus griseus* (Risso’s dolphin)	DD (it, 2013), DD (med, 2010), LC (gl, 2018)
*Ziphius cavirostris* (Cuvier’s beaked whale)	DD (it, 2013), DD (med, 2006), LC (gl, 2008)
*Monachus monachus*	DD (it, 2013), CR (med, 2008), EN (gl, 2015)

gl = global; med = Mediterranean; it = Italian. IUCN legend; DD: data deficient; LC: least concern; VU: vulnerable; NT: near threatened; EN: endangered.

**Table 5 animals-11-01663-t005:** Immunohistochemical (IHC) characterization and pulmonary location of ACE-2.

Species	IHC Results—LUNG Samples
*Ziphius cavirostris*	bronchiolar epithelial cells and alveolar macrophages +
*Tursiops truncatus*	bronchiolar epithelial cells +
*Stenella coeruleoalba*	bronchiolar epithelial cells and alveolar macrophages +
*Grampus griseus*	bronchiolar epithelial cells +
*Pseudorca crassidens*	bronchiolar epithelial cells and alveolar macrophages +
*Balenoptera physalus*	bronchiolar epithelial cells +
*Delphinus delphis*	bronchiolar epithelial cells +

**Table 6 animals-11-01663-t006:** Susceptibility of marine mammal species under study to SARS-CoV-2.

	Order	Species	IUCN Status	No. Non-Conservative Replacements	No. Residues Identical to Human	Mutations *	Predicted Susceptibility to SARS-CoV 2 Based on Amino Acidy Similarity
**Regular**	Cetacea	*B. physalus*	**VU** (med, 2010) (VU it, 2013)	1	20	**D30Q, Y41H, Q42E, L79I, M82T**	High
	Cetacea	*P. macrocephalus*	**EN** (med, 2006) (EN it, 2013)	2	22	**D30Q, L79T, M82T**	Medium
	Cetacea	*Z. cavirostris*	**DD** (med, 2006) (DD it, 2013)	2	20	**S19P, Q24K, D30Q, L79T, M82T**	Medium
	Cetacea	*G. melas*	**DD** (med, 2010) (DD it, 2013)	1	20	**Q24R, D30Q, H34R, L79I, M82T**	High
	Cetacea	*T. truncatus*	**VU** (med, 2009) (NT it, 2013)	1	20	**Q24R, D30Q, H34R, L79I, M82T**	High
	Cetacea	*Orcinus orca*	**DD** (med, 2019)	1	20	**Q24R, D30Q, H34R, L79I, M82T**	High
	Cetacea	*S. coeruleoalba*	**VU** (med, 2010) (LC it, 2013)	1	20	**Q24R, D30Q, H34R, L79I, M82T**	High
**Irregular**	Cetacea	*B. acutorostrata*	LC (gl 2018)	1	21	**D30Q, Q42R, L79I, M82T**	High
	Cetacea	*M. novaengliae*	LC (gl. 2018)	1	21	**D30Q, H34R, Q42E, M82T**	High

* In red we highlight non conservative replacement.

## Data Availability

The data presented in this study are available within the article and the Appendix A. The sequences generated in the present study were submitted to the GenBank database with the following accession number:MZ262275.

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
