# Peer review of "SARS-CoV-2, a Threat to Marine Mammals? A Study from Italian Seawaters"

_animals, 2021, doi:10.3390/ani11061663_

Round 1

Reviewer 1 Report

The paper by Audino et al. is well written and structured, and has a novel focus. The presence of SARS-CoV-2 in wastewater has far reaching potential implications for humans and, as the authors describe here, marine life. Clearly there is potential for SARS-CoV-2 or its RNA entering into marine waters, and the authors have done well to identify where this can occur and which species could be implicated by the impacts of this focussed around Italy.
I only have a few general comments, which mainly relate to the conclusions drawn from the data. Namely, whether what is being tested here is susceptibility to infection, or rather closeness at the sequence level supported by some IHC (which could inform subsequent studies, looking at infection). This can mainly be addressed in results or discussion, as suggested below, and there are a few minor points relating to presentation of tables and figures.

Figure 2-3 and related results: Title of Figure2-3 should perhaps describe what is being shown, rather than the sample. For those less familiar with the technique and aims, a short description indicating how the micrograph confirms the authors conclusions would be useful (line 400). Could the authors expand on line 401 (or in the discussion of this result) of how the finding is remarkable.

Table 3. The top 2 rows of numbers are confusing, could they be labelled? Could human sequence number be shown clearly. The colour legend shows grey but the table is light blue - presumably these are the same? Optional: rather than using colours to identify conservation, colouring by amino acid type (e.g. hydrophobic, polar) may reveal mutations that are likely to have a greater or lesser impact.

Table 4. Column titles would assist clarity.

Table 6 and related text.
In Table 6 the column showing mutations is not labelled.
The meaning of red highlighting M82T isn't clearly defined.
The final column in Table 6, regarding predicted susceptibility could be misleading: what the authors have evaluated is conservation and presence of ACE2, but susceptibility to infection can depend on many other factors (such as actions by proteases or other host factors, host immune responses). Likewise, risk to susceptibility to marine communities, more generally, will depend on the survivability of viruses in waste- and seawater (and reference 22 suggests a timeframe of a few days).
A more suitable way to summarise would hence be amino acid similarity or identity (%). It could then be discussed later what can be inferred from this, looking at the positions they have identified (e.g. 30, 82), if they are implicated in tightness of binding of ACE2 to spike protein. Currently this data is not linked in detail to the high/low susceptibility assessments; the manuscript would benefit from some expansion on this in the discussion.

Author Response

Point 1: Figure 2-3 and related results: Title of Figure2-3 should perhaps describe what is being shown, rather than the sample. For those less familiar with the technique and aims, a short description indicating how the micrograph confirms the authors conclusions would be useful (line 400). Could the authors expand on line 401 (or in the discussion of this result) of how the finding is remarkable.
R: We modified title and better described in the main text how the micrograph confirm our conclusion.

Point 2: Table 3. The top 2 rows of numbers are confusing, could they be labelled? Could human sequence number be shown clearly. The colour legend shows grey but the table is light blue - presumably these are the same? Optional: rather than using colours to identify conservation, colouring by amino acid type (e.g. hydrophobic, polar) may reveal mutations that are likely to have a greater or lesser impact.

R: Rows of number modified and labelled. We have corrected colour legend. Mutations that are likely to have a greater impact are non-conservative replacement (showed in yellow)

Point 3: Table 4. Column titles would assist clarity.

R: Done

Point 4: Table 6 and related text.
In Table 6 the column showing mutations is not labelled. R: correction done
The meaning of red highlighting M82T isn't clearly defined. R: We specified with * at the end of the table
The final column in Table 6, regarding predicted susceptibility could be misleading: what the authors have evaluated is conservation and presence of ACE2, but susceptibility to infection can depend on many other factors (such as actions by proteases or other host factors, host immune responses). Likewise, risk to susceptibility to marine communities, more generally, will depend on the survivability of viruses in waste- and seawater (and reference 22 suggests a timeframe of a few days).
A more suitable way to summarise would hence be amino acid similarity or identity (%). It could then be discussed later what can be inferred from this, looking at the positions they have identified (e.g. 30, 82), if they are implicated in tightness of binding of ACE2 to spike protein. Currently this data is not linked in detail to the high/low susceptibility assessments; the manuscript would benefit from some expansion on this in the discussion.

R: We specified in table that the predicted susceptibility to SARS-CoV-2 described in this table is based comparison between amino acid similarity of ACE2 sequence of each marine mammals with the human one. As regard implications of the position identified in tightness of binding of ACE 2 to spike protein, they are discussed at line 296-302.

Reviewer 2 Report

I have suggested some changes that may improve the manuscript.

Author Response

Point 1: Lines 27-30

“Abstract: zoonotically transmitted coronaviruses were responsible for three disease outbreaks since

2002, with the “Severe Acute Respiratory Syndrome Coronavirus-2” (SARS-CoV-2) causing the

dramatic “Coronavirus Disease-2019” (CoViD-19) pandemic, which affected public health,

economy, and society on a global scale.”

What about SARS-COV-1 and MERS? If you don't add them you don't need to say that they have

been three outbreaks, it is possible that they have been more than three, which could have been

undetected or under-reported. In fact, MERS occurs often on a year basis since it was discovered,

and these-days it seems that MERS is endemic in the Arabic region

(https://www.who.int/csr/don/01-february-2021-mers-saudi-arabia/en/) and maybe Sudan (doi:

10.3201/eid2512.190882).    

R: We deleted “three outbreaks diseases”, focus on only CoViD-19

Point 2: The English of lines 40 to 45 needs to be improved. For example:

“Furthermore, to evaluate if and how wastewater management in Italy may lead to susceptible marine mammal populations being exposed to the virus, geo-mapping data of wastewater plants, associated to the

identification of specific stretches of coast more exposed to extreme weather events, overlapped to

marine mammal population data, were carried out.”

to

Furthermore, geo-mapping data of wastewater plants was carried out to evaluate if wastewater

management in Italy may lead to susceptible marine mammal populations being exposed to SARS-CoV-2. In addition, specific stretches of coast more exposed to extreme weather events were overlapped to marine mammal population. (Or something similar. Short sentences are much clearer and concise than long ones).

R: Text improved: “Furthermore, to evaluate if and how Italian wastewater management and coastal exposure to extreme weather events may led to susceptible marine mammal populations being exposed to SARS-CoV-2, geomapping data were carried out and overlapped

I erase here “how”, since you don't indicate here “how”. I assume that you want to say that the

relationship between contaminated waters and mammals. This is in the next sentence. However, I

don't see the how. And I have the feeling that this will be the main problem of the manuscript.

In the abstract you don't say how it can be infected a marine mammal with a human Coronavirus,

and this will be probably the main deficiency of your manuscript. (See bellow)

It is known that there must be a close contact between humans and the host of SARS-COV-2 to be

infected, and that the main infection pathway from human to human, or another animal, is by the

air, mucus, urine or another corporal fluid (you can check this in the literature). From my point of

view, this means that the most plausible way of infection between marine mammals is through air,

mucus.,.. when groups of marine mammals are in close contact. I have to confess that I'm not

specialist in marine mammal epidemiology though. Then, I don't find a clear relationship between

swimming in waters with the potential of having Coronaviruses ( but see https://doi.org/10.3389/fmicb.2020.01795), and being infected with them. There is not a causal

relationship between both things, or at least a linear one.

R: We agree about the absence of a sentence about “how” in the abstract.

As expressed in the discussion (lines 482-486 and 499-501) we kept into account that reports of pathogens of terrestrial origin in marine mammals have already been observed, and that an exposure to untreated wastewater was deemed to be the possible source [38; 39].

Moreover, as reported in Mordecai et al 2020, despite the physical decay and loss of infectivity of SARS-CoV-2 at rates similar to other aquatic viruses, the risk to recreational users and fisheries  related to SARS-CoV-2 concentration in some coastal marine waters  is considered potential (e.g by filter feeding organisms or by onshore winds).

This aspect is also reported in Griffin et al 2003, where the spread of viral diseases through recreational water exposure and ingestion of contaminated shellfish is considered a public health concern, and it was shown that temperature, rainfall, type of disposal system and coastal processes such as tides and currents are responsible for the survival and transport of viruses in marine systems.

For the abstract, we modified the sentence accordingly as “Results showed the potential SARS-CoV-2 exposure for marine mammals inhabiting Italian coastal waters, putting them at risk when swim and feed in specific risk -areas.”

Introduction

Point 3: Lines 58 to 65.

Study [7] is a preprint. You can't use preprints to say that the virus is active in “water sources”.

What sources?? I have tried to find the DOI but it was not available with the number you give in the

manuscript.

R: We have updated the citation.

Point 4: Study [8] is a laboratory study with highly controlled conditions. In natural waters, particularly in

marine waters, there are lots of protists that can ingest many viruses per hour. It is highly possible

that many pathogens decay not only by the grazing of protists but also by the action of other

processes for example by the attachment to particles. I have the feeling that the survival of

coronavirus outside the different hosts must be very short in time, it is known to be from 2 to 4 days

in wastewater (https://link.springer.com/article/10.1007/s12560-008-9001-6). This could be

probably shorter in marine waters, for example Griffin et al. ( doi: 10.1128/CMR.16.1.129-

143.2003) has shown that pathogenic viruses may be detected by PCR in less than 10% of marine

waters, and being detected by PCR does not necessarily means that these viruses are active. Please,

you could read carefully the reference of Griffin et al. (2003), and others that you may find. Then,

you can discuss your findings in comparison to similar studies. In fact, if you read carefully the

reference of Rimoldi et at. 2020 [11 in your bibliography] it states in the abstract ...”Virus

infectivity was always null, indicating the natural decay of viral pathogenicity in time from

emission” So that I cannot perceive a direct relationship between being swimming in waters with

coronaviruses and being infected with them.

R: Thanks for the observation. We agree that study [8] is a laboratory study with highly controlled conditions, but we considered opportune to cite it as specifically referred to SARS-CoV-2 virus, not only to coronaviruses in general. Considering what explained in the paper of Mordecai et al 2020 we agree that the particle decay and loss of infectivity of the coronaviruses after arriving in aquatic habitats it is not the only aspect to consider, taking into account the risk of virus concentration e.g by filter feeding organisms or by onshore winds; moreover, as reported by Griffin et al 2003 about pathogenic human viruses in coastal waters, factors as precipitation, salinity and water temperature have been correlated with viral infectivity, representing some variables to take into account.

Point 4: Lines 80 and 85. I cannot see the direct relationship between “global warming and … the

occasionally flush of untreated or not efficiently treated sewage ...” Like this there is not link at all.

Global warming may change the raining regimes in Mediterranean ecosystems, which I think it is

uncertain (you have better to add a reference). But I cannot see a clear direct relationship, if there is

one, between higher rains and inefficiently treated sewage. Please, rephrase the sentence and clarify

this concept.

R: Increase of water temperature, heavy rains, floods and droughts will increase the distribution and patterns of human exposures to pathogens, chemicals and cyanobacteria; an inefficiently treated sewage makes human and animals more exposed to waterborne diseases.

We modified the sentence accordingly.

Further-more, it should be noted that extreme weather events associated to global warming (e.g. heavy rainfalls, storms, and floods), cause of sewer overflows, can play a relevant role in the coastal pollution ( Griffin et al 2003)

Point 5: Lines 88 and 96. I can see the point here. But, there is any study showing the potential of SARS

COV-2 infecting cells of marine mammals with ACE-2? If there is not you have one potential but

not one certainity. Because you have many coronaviruses in marine mammals, and then it is normal

that you can have these ACE-2 receptors. You have to discuss this and be clear, if you want to go

further with this study.

R: cited studies show how the risk of SARS-CoV-2 infection for cetacean is high because they are predicted to be highly susceptible to the virus according to similarity of ACE 2 sequence to human one.

Point 6: Table 1. I don't know how many Megaptera novaeangliae have you seen, I saw none. However, I

have been navigating the North Atlantic (Balaenoptera acutorostrata), the Central Atlantic

(Balaenoptera physalus), the Guinea Gulf (Physeter macrocephalus), the Mediterranean Sea

(Tursiops truncatus), and the Pacific (Delphinus delphis). Your list is obviously nicer and larger

than mine, but if you want to be realist, you have to work with field hard data not with lists. Check

the status of the different sea mammals populations in waters close to Italy, with abundances or

sightings and, then, make a new list with some numbers inside. In addition, if you have numbers of

affected mammals, you may add a second column, it would be fantastic!

R: the data in the table are the official ones of the Habitats Directive and it has been indicated that other data are available (mentioned in the discussions) but,for the purposes of the work, we have chosen to indicate only the Habitat Directive list.

As regards the number of specimens of "Italian" populations, there are data that only partially concern the Italian areas; we believe that adding numbers is not within the scope of the work and would not give an indication for all the waters around the peninsula.

There is the problem of climate change, which is very difficult to assess. You need to see if there is

a true risk: the probability times the consequences times the exposition. This is probably the section

I found more interesting. But you have to be more concise about risk assessment. How this risk

could change with changing conditions for example...  

Results:

Point 7:
267-268 I think that you are not “predicting”, you are assuming that if you have an acceptor you
have the potential to be infected. From my point of view, the concept is very different.

R: We modified with “suggested”.

Point 8: 270-276. See the previous comment about the Table 1. I do not know what regular or irregular
means in terms of abundances. 277-280. If you only have 8, you don't have to care about the others
in your list, isn't it? Or if you have because you find them you can clearly say that this is novel, and
this is one important point of your study!

R: Regular or irregular is not related to the individual numbers but to the reported presences and range. We can have a regular specie and no information on the abundances.

1) regular species - species which occur regularly in the region;

2) irregular species – species which do not have a stable and/or regular occurrence in the biogeographical/marine region; and for which the number of records is insignificant. Categories are explained by the explanatory notes and guidelines (file:///C:/Users/lauriano/AppData/Local/Temp/Reporting%20guidelines%20Article%2017%20final%20May%202017(0).pdf) see also https://nature-art17.eionet.europa.eu/article17/species/progress/?period=5&group=Mammals&conclusion=population Moreover to be consistent with the habitat directive the names regular will be corrected in present regularly and irregular in occasional.

We have 15 but some are regularly present; Megaptera is irregular as well as the other two listed in the table 1. To this extent for example in these days a grey whale has been reported along italian coast but this can be reported as sporadic (less than irregular)

Point 9: 301-302 If you are talking about Italy it is not necessary to talk also about Gibraltar Strait; focus,
focus, focus.

R: yes you are right; the Gibraltar Strait mention was to higlight a different evaluation at local scale. We delete it.

Point 10: Immunohistochemical probe
This section is very interesting, but if besides SARS-COV-2 Coronaviruses are found in marine
mammals, you have to state if your immunochemical detection is specific for SAR-COV-2 or it has
a broader detection that may be related to other Coronaviruses. This is very important, because if
you have a probe that detects only receptors for SARS-COV-2, then you are closer to say that there
can be an infection and maybe an intermediate host, than if your probe may only detect a receptor
against any kind of Coronaviruses.

R: We would like to thank the Reviewer for these highly valuable and precious comments and remarks on her/his behalf. In this respect, albeit playing a key role in host-pathogen interaction at the cellular level, ACE-2 does not appear to be the only receptor for SARS-CoV-2. Indeed, additional cell receptors, either co-expressed or not concurrently expressed with ACE-2, have been reported to be involved in viral-host cell interaction. This is the case, for instance, of "neuropilin-1" (NP-1)  as well as of the "high-density lipoprotein (HDL)-cholesterol scavenger receptor B-type 1.

Furthermore, to the best of our knowledge, the ACE-2 molecule is known to serve as the main cell receptor for SARS-CoV-2 as well as for SARS-CoV beta-coronaviruses, while MERS-CoV has been reported to use CD-26 (or dipeptidyl-peptidase-4) as its main cell receptor".

Discussion
Point 11:
451-454 You don't have shown this. You have find that they have a receptor, that they can have
Coronaviruses, and that they might have the potential to get a Coronavirus like virus. But if you
don't have find SARS-COV-2 inside their lungs, you are not sure about this. Clarify this better and
be careful with your language.

R: we modified with “potentially highly susceptible to the virus”

Point 12: 470-473 Read the comment about this I gave above.
484-492 I find that there is a problem of risk quantification. You have to be more clear about this
risk and how many risk do they have.

R: as dated in the manuscript it is always a potentially risk

Point 13: 512-522 I have the feeling that this is the main way of transmission of viruses between marine
mammals, and from my point of view this is much more risky than swimming in waters with
Coronaviruses. If it happens as in humans, you can go walking by the street without problems, but if
you are in close contact to someone that has SARS-COV-2 or within a group that is infected, then
your odds will be higher.

R: we talked about their gregarious behavior as route of transmission and diffusion of virus between cetaceans; this is not correlate with the risk for a cetacean to come into contact with the virus if introduced into marine waters from contaminated water courses, and then, in the case of infection, of spread it.

Point 14:
537-548 This section is not good at all. We expect to end this pandemic before the end of the year.
Once finished the odds of one marine mammal to be infected with human coronavirus will be very
small if they will not be a reservoir. I have the feeling that this is very improbable. You could check
for this in a future by doing PCRs of coronaviruses in marine mammals. However, I think that you
have to check also to other coronaviruses and viruses pathogenic for marine mammals. You odds to
find a positive result will increase.

R: We are grateful to the Reviewer also for these additional useful comments and remarks on her/his behalf. In this regard, it is not possible, at present, to precisely define a timeframe within which the SARS-CoV-2 pandemic will come to an end, given the many host-, virus- and environment-related variables that are in play, along with their mutual interrelationships. A key player within such and intricate and challenging context will be undoubtedly represented by the achievement of an ad hoc "herd immunity level" conferred by the mass vaccinations against CoViD-19, which should be reached not only at a national and/or continental level but, most importantly, at a global level.

In the meantime, taking also into adequate account that SARS-CoV-2 most likely is a zoonotic pathogen, for which clear-cut evidence of susceptibility has been reported in a range of domestic and wild mammalian species under both natural and experimental conditions (Shi et al., 2020, Science; Di Guardo, 2020, J Comp Pathol), it seems reasonable and biologically plausible to believe that, once SARS-CoV-2 should be "put under control" within the human population thanks to the aforementioned mass vaccination campaigns, followed by the obtainment of an ad hoc "global herd immunity level", the virus could naturally "jump" (so-called "spillover") into susceptible (and non-vaccinated, immunologically naive) animals, thereby giving rise to new, scary "variants of concern" (VOC), as clearly shown in spring and summer 2020 in intensely reared minks from The Netherlands and Denmark, who caught SARS-CoV-2 infection from their breeders, owners and/or caregivers and in whose organism the so-called "cluster 5 VOC" was generated, to be subsequently transmitted to the aforementioned people (so-called "spillback").

Within this context, it should be additionally underscored that a domestic cat from North-Western Italy (Piedmont Region) was recently found to be infected by the SARS-CoV-2 "English"(alias "B.1.1.7") VOC, which was most likely acquired from its CoViD-19-affected owners (IZSPLVA, 2021).

As reported in our manuscript, the high homology levels existing in the ACE-2 receptor molecule from the herein investigated cetaceans as compared to the human one, would make such a worrisome "scenario" possible/plausible also for sea and ocean ecosystems. This is additionally emphasized by the high and progressively expanding levels of marine (and terrestrial) environmental contamination on behalf of disposable face masks, an undefined percentage of which could still harbour viable SARS-CoV-2 (with which free-ranging cetaceans could come into contact), coupled with the repeatedly documented and consistent viral excretion by the faecal route on behalf of SARS-CoV-2-infected patients. As a matter of fact, based upon the results of a study carried out by Chinese Colleagues, approximately 60% of CoViD-19-affected individuals would shed through their faeces SARS-CoV-2 for a median time of 22 days (British Medical Journal, 2020)".

Round 2

Reviewer 2 Report

Nice work.